# The Impact of Social Well-Being on Population Diet Nutritional Value and Antiradical Status

**DOI:** 10.3390/foods12132619

**Published:** 2023-07-06

**Authors:** Victor Gorbachev, Igor Nikitin, Daria Velina, Natalia Zhuchenko, Alexander N. Kosenkov, Andrey Sokolov, Igor Zavalishin, Alla Stolyarova, Evgeny Nikulchev

**Affiliations:** 1Research Laboratory of Nutritional Systems Biotechnology, The Plekhanov Russian University of Economics, 36 Stremyanny Per., 117997 Moscow, Russia; foodstudy@yandex.ru (V.G.); kattim67@gmail.com (D.V.); 2Department of Biotechnology of Food Products from Plant and Animal Raw Materials, K.G. Razumovsky Moscow State University of Technologies and Management (the First Cossack University), 73 Zemlyanoy Val, 109004 Moscow, Russia; 3Department of Medical Genetics, I.M. Sechenov First Moscow State Medical University, 8-2 Trubetskaya Str., 119435 Moscow, Russia; zhuchenko_n_a@staff.sechenov.ru; 4Department of Hospital Surgery, I.M. Sechenov First Moscow State Medical University, 8-2 Trubetskaya Str., 119435 Moscow, Russia; alenkos@rambler.ru; 5Mental-Health Clinic No. 1 Named after N.A. Alexeev, 2 Zagorodnoe Shosse, 117152 Moscow, Russia; azaleptine@yandex.ru; 6Higher School of Public Administration, Financial University under the Government of the Russian Federation, 49 Leningradsky Prospekt, 125167 Moscow, Russia; ivzavalishin@fa.ru; 7Department of Management and Economics, State University of Humanities and Social Studies, 30 St. Zelenaya, 140400 Kolomna, Russia; stolyarova2011@mail.ru; 8Department of Digital Data Processing Technologies, MIREA—Russian Technological University, 78 Vernadsky Avenue, 119454 Moscow, Russia

**Keywords:** antiradical potential of diets, eating habits of consumers, “Fast food”, epidemiological properties of food, social stratification, quality of human life, antioxidants, vitamins, nutritional balance, sociological survey

## Abstract

The paper presents the result of assessing the antiradical status of consumers (in the context of Russia) in connection with their well-being. This approach is based on a multistage study, in which the results of sociological surveys were applied, as well as estimates of the antiradical potential (ARP) of diets obtained using neural networks, bootstrapping the chemical composition of diets, and calculating reference values using mathematical models. The paper presents data collected from residents living in the territories of at least 21 regions and cities of Russia: Magadan, Saint Petersburg, Moscow, Krasnodar, Lipetsk, Vladivostok, Novosibirsk, Omsk, Voronezh, etc. A total of 1001 people were interviewed, which, according to our calculations, gives a margin of error in value of approximately 3.1%. To calculate the lack of vitamins in the diets of residents of the Russian Federation, data on the chemical composition of food products from the FNDDS database were used. The assessment of dietary habits showed a lack of vitamins below the recommended level in 73% of Russians for vitamin D, 59% for retinol, 38% for β-carotenes, 13% for vitamin E, and 6% for ascorbic acid. The study showed that at least 36% of the Russian population has a low antiradical status, while it was found that “poor” consumers are more likely to consume economically more expensive foods (in terms of their nutritional value). The “poor” segments of the population consume 180–305% more canned food and 38–68% more sweet carbonated drinks than other social groups, but their consumption of vegetables is 23–48% lower. On the contrary, “wealthy” consumers consume 17–25% more complex (varied) dishes, 10–68% more fresh vegetables, and 8–39% more fish. From the obtained values it follows that consumers with low levels of ARP in their diets are in a group with an increased probability of a number of “excess” diseases (diseases of the cardiovascular system, obesity, etc.). In general, the ARP values of food consumed for low-income segments of the population were 2.3 times lower (the ratio was calculated as the percentage of consumers below the level of 11,067 equivalents necessary for the disposal of free radicals generated in the human body per day) than for those who can afford expensive food (consumers with high income). A simple increase in consumption of unbalanced foods, in our opinion, will only contribute to the entry of these consumers into the “average diet trap”. All this makes it imperative to develop comprehensive measures to create a new concept of public catering; otherwise, we can expect a reduction in both the health of the population and the performance of the economy of the whole country.

## 1. Introduction

Since about the middle of the last century, there have been significant changes in the production of agricultural raw materials. These changes, the active phase of which took place in the 1960s, were later called the “Green Revolution” [1]. These changes have led to a dramatic shift in the public catering of many segments of the population from around the world, namely an increase in the proportion of foods with a high content of sugars, vegetable and animal fats, saturated fatty acids, as well as animal products in general [2]. Such availability of food raw materials contributed to the saturation of consumer markets with appropriate food products and the spread of such alimentary-dependent diseases, including type 2 diabetes, cardiovascular diseases, autoimmune intolerances to certain types of products, obesity, etc. [2,3,4]. In addition to the above, the intensification of production also increased the anthropogenic impact on the environment and expanded the use of a number of agricultural xenobiotics [5].

Recent studies presented by the European Society of Cardiology (ESC) in 2019, including results from 56 countries, have identified a number of problematic points [6,7,8]. The first point requiring attention is the fact of the uneven distribution of this group of diseases across the world. The second is that over the past 30 years, there has been a 2–3-fold increase in the prevalence of obesity and diabetes (world average), which has made it unlikely that the goals set by the WHO to stop the growth of these risk factors by 2025 to be achieved [8].

Similar studies have shown that all countries since about 1990, faced with the problem of increasing cardiovascular diseases and obesity in their populations, are divided into three groups according to the level of their socio-economic development [9]. At the same time, developed countries over the past 20 years (1990–2010) have been able to develop social measures to prevent the growth of these diseases in disease structures of these countries, which is not always true for economically underdeveloped and developing countries [8,9].

As already mentioned above, a number of economically developed countries have taken measures since the 1990s to reduce the exposure of their citizens to diet-related risk factors [9]. As was rightly noted, poor and economically underdeveloped countries have mostly struggled with infectious diseases and famine in their territories. Economically developed societies, on the contrary, have realized a possible problem with excessive food consumption [9].

Developing countries, on the one hand, have already received a sufficient amount of resources and have reached the minimum limits of food security, and on the other hand, they have not carried out a significant revision of their attitude to their public catering policy [2,7,9,10]. Thus, it can be expected that more often than not, the results of advances in food security in developing countries have led to an increase in the caloric content of diets but not to an increase in their biological value and balance.

It is necessary to take into account the fact that the nutritional value of foodstuffs has been decreasing for at least the last 50 years. First of all, this is due to an increase in carbohydrates and a decrease in alimentary factors such as ascorbic acid, niacin, and other nutrients in the quantitative range of ≈10–50% for various substances [11].

Also, to date, data have been collected on a direct relationship between an excess of alimentary factors (caloric content of food, and sugar or fat content) and free radical cardiovascular pathologies induced by them, as well as pathologies associated with diabetes mellitus, etc. [7,8,9,10,12,13,14,15,16,17,18].

Recent nutritional data points to the need to limit saturated fat intake. However, food products in which saturated fats are replaced by refined carbohydrates and, in particular, added sugars (for example, corn syrup containing fructose) become no less of a problem [19].

Studies show that consumers who eat diets high in sugar may be more likely to develop coronary heart disease (almost 300% of the normal rate) as well as metabolic syndrome with elevated glucose, uric acid, or insulin and leptin resistance, and as a consequence, non-alcoholic fatty liver disease may also develop (the latter currently affects ≈25% of people worldwide) [19,20,21,22]. Fatty degenerations have been shown to be closely correlated with other diseases, such as type 2 diabetes mellitus [20,21].

Research in this area has been ongoing for a long time—since about the mid-1970s—and it has been known that the global pandemic of non-communicable diseases lies in part in the diet of radical-generating “over-processed” foods. Despite this, society has been slow to initiate any intervention, partly due to the lobbying of interests of the “food industry” [23], so some authors have called for the control of the highly processed food market.

In addition to the above, modern unbalanced diets can indirectly affect the immune system through the human gut microbiome [20,24] and induce a number of pathogenic processes through the generation of free radicals in the human body [12,13,15,17,22]. So, excessive consumption of unbalanced foods leads to excessive generation of radicals, which makes it necessary to increase the ARP of foods.

As we have shown earlier, the antiradical activity of food products must be taken into account when assessing the balance of diets, and in general, this parameter can be regarded as one of the complex measures of food quality [17].

All this makes it possible to use components with antiradical action for the development of new diets that prevent free-radical pathologies [25]. On the other hand, in order to develop any diet, it is necessary to analyze the existing antiradical status of the population of a particular country by mirroring with how it was previously achieved for certain groups, for example, for vegetarians in Finland [26]. However, there are challenges in collecting dietary data at the population level. Thus, multiple studies in food epidemiology have shown that the method of collecting data in the form of point statistical estimates is not without error [27,28,29].

Similarly, it can be expected that point estimates of the antiradical status of the population will also contain this statistical flaw. While anticipating this possibility, in our previous studies on the assessment of the antiradical potential (ARP), we used interval values instead of point values. In our opinion, this has made it possible to avoid a number of statistical limitations [17,25]. This parameter was chosen by us as a kind of “collective complex value of nutritional value”, depending both on the balance of the chemical composition of food and the terms and conditions of its storage on the one hand [17], and it can affect the course of free-radical human pathologies on the other hand [12,18,26].

It is known that the social status of a person and their well-being affects the nature of the diet that a person prefers or is forced to adhere to [30]. However, not all researchers consider the factor of social stratification as significant in the formation of food rations [31].

For this reason, the goal that we set before this study was to assess the impact of social well-being on the balance of the ARP diet of an average consumer using the example of Russia, including understanding the pathogenicity of the diet to predict its impact on the health of the human population in future.

## 2. Materials and Methods

Three groups of data served as materials for this study. Some introductory data types have already been published by us earlier [17,25,32]: 1. data on habitual diets and the frequency of consumption of certain food products (this study); 2. ARP values of food products from a large number of food groups and changes in food redox potentials with various processing methods [17,25]; and 3. data on the reference levels of the minimum volumes of substances capable of utilizing free radicals produced in the human body [32].

The generalized scheme of the experiment for this study is shown in Figure 1.

The collection of sociological data was carried out by several methods for a total duration of 3 months and took place from 14 December 2021 to 14 March 2022. Some of these results were obtained from respondents through a personal survey (54 people) in Krasnodar Territory and Magadan Region (Russia). If it was impossible to conduct a personal survey, printed forms were distributed (85 people in the Magadan region, the forms were filled in by those respondents and returned later). These two territories were taken as almost two extreme points, between which the predominant part of the territory of Russia is located. The first is located in northeast of Asia (near sea of Okhotsk), and the second is located in the south-east of Europe (near the Black Sea). Such an approach, in our opinion, could guarantee the greatest coverage of data on social groups and places of residence in Russia.

A Google form for an online survey was also created, and a link was sent to respondents (using instant messengers and social networks) living in one of the 21 provinces and large cities: Saint Petersburg, Leningrad Region, Moscow, Moscow Region, Krasnodar Territory, Lipetsk Region, Primorye and Vladivostok, Republic of Sakha Yakutia, Novosibirsk Region, Omsk Region, Penza Region, Khabarovsk Territory, Ivanovo Region, Altai Territory, Perm Region, Republic of Tatarstan, Sverdlovsk Region, Voronezh Region, and Karachay-Cherkessia. These regions are located in different geographical areas of Russia, with different levels of prosperity and the ability to grow their own food and, as a result, with different eating habits of the residents living in them. The population of these regions is approximately 52.6 million people, corresponding to 35.8% of the total population of the country. Respondents were also asked to involve people with whom they are directly acquainted (relatives, work colleagues) in this study, which increased the geographical coverage, and it can be expected that residents in a majority of the regions (provinces) of Russia took part in the data collection. In total, 862 people were interviewed using the electronic form. A total of 1001 people took part in the study.

Data collection by different methods (survey via electronic form and direct interviewing) was dictated by necessity, since different age groups have different computer literacy skills and some groups of older people, including from remote areas of the country (rural population from Magadan and Krasnodar regions) would not have been interviewed for research purposes if only an electronic form had been used.

The link to participate in the sociological survey (invitation) was sent to ≈8436 people, but the number of those who agreed to give answers was approximately 10.2%. As of 2021, more than 146 million people live in Russia, which, when calculating the representativeness of the sample at 95% CI, gives an estimate of ≈3.11% of the margin of error of the values for the sample in this study, if calculated for the entire population of the country.

The survey form was identical in content to the printed forms and questions asked during the personal interview. Among these questions, there were standard ones, as well as those about the preferential choice of food products in accordance with their price category (well-being of respondents). The questionnaire for the survey (form) was written in the Russian language, but we attached a translated version of it as Appendix B for this article.

The main block consisted of 17 questions that were devoted to various aspects of nutrition and diet selection by Russian citizens (food habits). They clarified how often respondents eat certain food groups, such as dairy products, meat products, fruits and vegetables and their processed products, fish and fish products, bread and bakery products, eggs and their processed products, cereals, sweet carbonated beverages, juice drinks, confectionery, and canned foods. It was also specified how often respondents visit bistro-type eateries (fast food) and how often they consume complex culinary dishes (soups, casseroles, etc.). For each question following the format of “how often do you eat … this or that type of food?”, 5 answers were given: 1. Daily; 2. I eat it every 2–3 days; 3. I eat it once a week; 4. I eat it less than once a week; and 5. I practically do not eat it.

A total of 17 food groups were evaluated, which substantially corresponded to similar food groups in the questionnaire for the sociological survey. For each food group, the following products taken from the database of Food and Nutrients for Dietary Research (FNDDS) [33] are evaluated: 1. Dairy products; 2. Bread products; 3. Meat products; 4. Cereals; 5. Confectionery; 6. Complex dishes; 7. Processed vegetables; 8. Fresh fruits; 9. Fresh vegetables; 10. Eggs; 11. Processed fruits; 12. Juices; 13. Carbonated sweet drinks; 14. Semi-finished products; 15. Canned food; 16. Menu from bistros (fast food eateries); and 17. Fish.

The expanded list of food products used in this study for statistical analysis is presented as Appendix A in this article. A total of 1315 items were analyzed, and the data were published earlier [25].

The results obtained from data collection were processed in Excel 2010, statistical tests (including bootstrapping and dendrogram construction) were performed in the Past program, and graphic materials were built using the Scimago Graphica and Corel Draw programs.

On the basis of the obtained data on the frequencies of preferred food groups, a simulation was made of the balance of the diets of Russians in terms of their restorative capacity. The calculation of such values was carried out using the bootstrap method (weighted average) in Excel 2010 with 5 thousand iterations. The food compositions for comparison were taken from the FNDDS database [33], and the reasons for choosing the FNDDS database were indicated by us earlier [25]. We used bootstrapping methods to evaluate statistical parameters, since we did not have an initial assumption about the nature of preferences of each of the participants in this study (unbiased model). In this case, we can estimate the average values and the spread of these data for the entire population, and bootstrap procedures are the best suited for this.

This approach allowed us to simulate a random choice of food products by consumers with the following parameters: the mass of food consumed was calculated based on the recommended average value for dry weight and moisture content of 68.3% (average value for our data from FNDDS). The norms of food components recommended for daily consumption in Russia and the United States are quite similar (Table 5 from [25]), we have previously evaluated the ARP of each of the accepted norms using neural networks [25].

According to these recommendations for daily food intake, the mass of main food components for the adult population ranged from 510 to 1046 per day (an average of ≈778 g, regardless of gender and profession) or, taking into account humidity, varied from 1608 to 3299 g (in an average of 2454 g), which averaged 15 positions for bootstrapping (out of 17 food groups we analyzed). Thus, the average serving size from each food group was 159 g in the analysis.

As we have already noted, the reference values were selected. They were calculated by setting the lower statistical threshold at 5310.4 equivalents (obtained by adding the equivalents required for the utilization of xenobiotics in 1222.7 and 4087.7 from a previously published study [32]). This value was chosen because neither the gender nor the habits of the consumer for whom the ARP was evaluated were known in advance. All values below this limit were accepted by us as obviously pathogenic.

The second reference value is indicated at ≈7967 equivalents. It was obtained as the average of the minimum estimates of the 95% confidence interval (95% CI) of the ARP of the daily recommended intake using the ANN [25]. All values in the range from 5310.4 to ≈7967 will be designated as propathogenic, since under certain conditions they can contribute to the development of pathologies.

The third reference value is indicated at the level of ≈9516 equivalents. It was obtained as the average value of the ARP of the daily recommended intake using the ANN according to the Codex Alimentarius and recommendations from national academies of sciences of Russia, the USA, and the WHO [25]. If the values of the ARP estimates fall in the range from ≈7967 equivalents to ≈9516 equivalents, then such a diet will be called unbalanced in terms of its redox ability.

The fourth value is indicated at the level of 11,067 equivalents as the average value of maximum estimates of 95% CI ARP of the daily recommended intake using the ANN according to the Codex Alimentarius and recommendations of national academies of sciences of Russia, the USA and the WHO [25]. If the estimates of the ARP of a diet fall in the range from 9516 equivalents to 11,067, such a diet will be called poorly balanced.

If a value greater than 11,067 equivalents was obtained, then the corresponding diet was given the status of balanced in terms of its redox ability.

## 3. Results and Discussion

Let us analyze the collected sociological data. Figure 2 shows the level of well-being of the respondents. Most of them (slightly more than 2/3) indicated that they mainly buy products from the middle price category. Almost equally (≈6%), there were votes from participants who often prefer to purchase expensive food products and those who do not always have enough money to buy basic products. At the same time, every fifth respondent indicated that he buys only basic economy class products (bread, cereals, potatoes, etc.).

It should be noted that percentages obtained for the poorest and richest representatives of society are close to percentages obtained for surveys conducted in Poland [34]. The rest of groups are different.

Let us point out immediately that in relation to those respondents who gave answers about the predominant purchase of expensive food products, no age, gender or professional connection was found. It is therefore not entirely fair to the group of respondents who indicated that they are not always able to buy basic foodstuffs. The last group almost twice as often included the unemployed, pensioners, and blue-collar workers (approximately 12, 8, and 23%, respectively). The ratio of professions is not shown.

Based on the data obtained on 17 food groups, an analysis was made of their shares in the generalized diet. Statistically speaking, these are essentially the probabilities that consumers will choose one or another representative from each named food group in their diet. The values are presented in Table 1.

As can be seen, bakery products and meat products make up a fairly large share in diets of Russian residents. They account for about 1/5 of the total volume. The five most popular food groups account for approximately 47% of the total diet of Russian consumers. It is for these groups of products that low ARP values were found (with the exception of complex dishes, which may vary in this parameter). It suggests a weak balance in the reducing ability of the diet in relation to free radical particles [17,25]. However, the data in the table represents average values for the entire sample. Based on the responses received, it was assumed that there are significantly diverging values of the ARP of diets for poor and rich segments of the population, especially taking into account the latest data published by us earlier [25,32].

To begin with, let us model the diets of people in the different levels of their material well-being (Figure 2), for each group separately. During the simulation, significant deviations from the average values were revealed (Table 1). The results of testing this hypothesis are shown in Figure 3.

As we can see in figure above, respondents who answered that they could not always buy “basic products” (poor consumers—green lines, Figure 3) most often indicated that they prefer products that do not require wasting time such as “Fast food”; products with a long shelf life (canned food); instant products, such as custard noodles (analogue of “Asian cuisine”) or convenience foods; sweet sodas; and juices and nectars, and also consumed more confectionery products than average.

To a certain extent, a similar choice pattern is revealed for consumers who answered that they buy mainly “basic class products” (blue lines, Figure 3). In accordance with the identified distribution pattern, consumers from this category located in the middle between the “middle price class” (violet lines, Figure 3) and “poor consumers”. A high Pearson correlation was found between the responses received from these two consumer groups (r = 0.91 at *p* < 0.001) (green–blue lines). Immediately, we note that a smaller data correlation was found between the “poor” and the “middle class” (r = 0.85; violet–green), and an even less correlation between the “poor” with the “rich” class (r = 0.75; orange–green lines, both of the latter are significant at *p* < 0.001).

The application of the nonparametric Fligner-Killeen test for the homogeneity of group variances between the “rich” and “poor” groups showed statistically significant differences (*p* ≈ 0.035). This test did not reveal them when comparing between the “poor” consumers and consumers of “economy” class products (*p* ≈ 0.18).

The average consumer who prefers “middle price products” consumes the first seven categories of goods (numbered 1–7, Table 1) at below average levels. These consumers do not differ from the average values, because they themselves set the main consumption trend in the general population of people surveyed.

The distribution of food consumption patterns between consumers who prefer “expensive food products” and other groups varies greatly. They are most likely to consume fish and fish products, fresh as well as processed vegetables, and, most importantly, complex meals. At the very least, the values of consumption of complex dishes for this category of consumers are higher than for all others, so it can be assumed that these people have the financial opportunity or free time to cook for themselves (or employ a personal chef). They consume less bread, dairy products, confectionery, as well as juice products and eggs than others. In Russia, products of this type most often contain additional added sugar (fast carbohydrates, the concentration of which can reach up to 10 g or more per 100 g of product).

Tukey’s test also showed differences between two groups of responses, “rich-poor” (*p* ≤ 0.001), and according to the same test, there were no differences between the groups of “economy class” and “poor” consumers (*p* ≥ 0.1). Statistically significant differences were found between consumers of “economy” and “rich” products (*p* ≈ 0.005). The probability of deviation of H_0_ in this case is quite high, however, these values are still below the Bonferroni correction ≈0.008, which we applied for multiple comparisons.

All this cannot but affect the antiradical status of each of consumer groups. Let us also note a very important relation between the incorrect choice of food consumers and their low level of well-being. Surprisingly, however, it has been confirmed that the behavior associated with the preference for foods with low restorative potential can also be associated with the unwillingness of “poor” consumers to wait.

Indeed, by now, the results of the “marshmallow” experiments devoted to “delayed pleasure”, conducted in the 1960s, are known. Reanalysis of this data showed that one of the important reasons is the person’s social status. So children from poor families more often did not want to wait, but preferred to receive rewards immediately [35].

This data is similarly consistent with results we obtained at the stage of sociological research—“poor” consumers preferred “food without time costs” (Figure 3). However, it is this approach that harms them to a greater extent, since products from these categories (carbonated water, Fast food, convenience foods, canned food, etc.) have low ARP values. And low consumption of fresh vegetables (food category 9 in Figure 3) worsens the recovery status of diets of this group at the same time.

We also note that in terms of 100 g of nutrients, the products of these categories (numbered 1–3 in Figure 3) are “not cheap”. For example, consuming of 100 g of protein would require ≈2.5 pieces of rye-wheat bread or ≈15.5 servings (90 g) of custard noodles, which would give a price difference of ≈7.9 times in favor of bread. Therefore, the choice of this category of goods by consumers is not economically feasible and affects them negatively not only from the point of view of worsening their antiradical status, but also in regard to their budget, and even contributes to an imbalance in nutritional properties of their diets. It could be assumed that bakery products are little consumed by this group of “poor” consumers due to intolerance to bread protein (celiac disease), but this assumption does not stand up to criticism, since noodles also contain the same protein (in Russia, it is most often made from wheat flour). To adhere to their budget, it is recommended that this group change their strategy for choosing food.

In addition to the above discussion, let us estimate the percentage of consumers whose habitual diet may be below ARP values for the diets recommended by the WHO, the Codex Alimentarius, and the national Academies of Sciences of the USA and Russia [25]. The bootstrap values are shown in Figure 4.

We point out right away that the ARP confidence estimates during their bootstrapping for all four consumer groups using the Tukey test differ from each other, both in terms of an average value parameter and in the nature of the equality of dispersion distributions (*p* ≤ 0.0001 at 99.9 thousand recalculations).

Taking into account the data presented, we estimated the percentage of people in total for the entire general population. It averages ≈1% of consumers whose dietary assessments are in the pathogenic range, 10% of which fall into the propathogenic range, ≈15% of unbalanced diets, and about 20% of poorly balanced ones, with slightly more than 53% of all respondents have a balanced diet, according to our estimates. These figures, however, may vary depending on the ongoing social processes in society.

Thus, people from the group “who prefer buying cheap food” (canned and fast food) risk their diet falling almost twice as often into the area of pathogenic values (area 1 in Figure 4, less than 5310 equivalents). Upon further comparison, it is revealed that the total probability of falling into the propathogenic and unbalanced areas of dietary assessments of ARP for “poor” consumers is ≈3 times more likely than for those who purchase “expensive food products”. In general, an unsatisfactory situation can be recognized for ≈72% of consumers from this low-income group (estimates below 11,067 equivalents), which does not exceed half for other consumer groups and less than 1/3 (≈32%) for “rich” consumers.

Methodically, it is possible to estimate the percentage of consumers with reduced consumption of a number of vitamins (it is difficult to do this methodologically for each group). However, if we apply the bootstrapping algorithm for the entire general population, while considering how it was achieved with the ARP values and at the same time borrowing the concentration of vitamins from the FNDDS database [33], then we obtain a series of values. They are presented in Table 2.

We point out that the given values for the recommended daily intake in Russia are generally close (Russia, https://www.garant.ru/products/ipo/prime/doc/402716140/ access date 3 July 2023) and [25] to the values proposed by the National Institutes of Health (USA, Bethesda, MD, https://ods.od.nih.gov/HealthInformation/dailyvalues.aspx access date 3 July 2023).

The first thing that can be immediately revealed when analyzing Table 2 is the fact that even when calculating using databases on the chemical composition of food products, a rather high percentage of some scarce components is revealed among Russian consumers. This may also explain the insufficiently satisfactory ARP values that we have identified. These vitamins (not taking into account the polyphilic components of plants) are often attributed to the main ARP of the food systems that a person eats [18,36,37,38]. If we assume that chemical compositions presented in the FNDDS database objectively correspond to the chemical composition of products, then the diet of the average consumer is depleted in a number of indicators. This ultimately affects the deterioration of population health [39,40,41]. From the given values, it can be seen that the greatest deficiency is observed in vitamin D. Approximately one in ten is deficient in vitamin E, and one in twenty is deficient in vitamin C and folic acid. Diets for vitamins A and β-carotene appear to be even less balanced.

As for the limitations of this study. We would attribute two types of difficulties to the disadvantages of this sociological study. 1. Our sample is not as large as we would like. We have not covered all geographical regions of Russia in this study, and therefore the conclusions we have obtained can be compared with other data (with other researches studies) with some caution and reservations. The data obtained by us were collected while taking into account the approximate metric estimates and, for this reason, contain a fairly wide range of values comparable to about 3.1% for each value. The obtained values, however, are comparable by the spread of values (confidence intervals 1-α, where the last value is 5%).

## 4. Comparison with Published Data from Other Countries of the World

Currently, big data has been accumulated on the relationship between a person’s diet and their socio-economic status (SES), including their monetary income [42]. Also the relationship is shown between the presence or absence of crisis events taking place in society and the levels of consumption of a large amount of high-calorie foods, such as fast food, including due to the psychological stress experienced by the consumer [42]. Moreover, an inverse correlation of r = −0.4 was found between the unemployment rate and the price index for fruits and vegetables relative to the price of purchased dishes from restaurants, including fast food establishments [43]. One of the reasons for the increase in the price of fast food can be associated with the increasing demand for this type of food [42,43]. Moreover, as shown in studies conducted earlier in Russia, a large proportion of fast food is consumed by people with a lack of time, as well as money (mainly working in several jobs at the same time) [44]. Therefore, as we noted above, referring to marshmallow studies, low-income people show less patience and discipline in relation to their eating behavior [35], and the term fast food is the best description of the current situation. Fast food has been shown to be readily available and quickly prepared, and working adults may prefer it to other meals at mealtimes [44], with little awareness of the future consequences of overconsumption. At the same time, our data shows that consumers rather do not adequately evaluate the price–quality ratio per unit of the nutritional value of the product, since canned food, and equally fast food products, is not at all cheap in terms of their nutritional value, which may indicate a lack of economic literacy of consumers in this matter.

Despite this, researchers have attempted to describe the increase in demand for “unhealthy” food as a function of market prices, income, time and preferences of consumers, and the availability of fast food restaurants or appropriate places to buy fast food [44,45,46]. However, the sociopsychological aspect is not excluded. It is associated with the education and upbringing of a person in their early periods of life, with an understanding of happiness and the habit of engaging in self-reflection [47]. We believe that no less important is the habit of self-discipline, which affects, among other things, the ability to prepare food for oneself [44,48]. We do not have direct ways to compare the antiradical status presented in our data and data published by other researchers, so we will focus only on certain aspects of changing food preferences depending on the well-being of consumers from certain countries as an example.

One of the generalizing studies conducted considering the data from 10 countries, Canada, New Zealand, France, Spain, USA, Holland, Sweden, Italy, etc., shows the difference in prices between a serving of healthier and less healthy forms of food [49]. And if the differences were identified for meat and grain foods, as they could not be identified for items such as sweet sparkling water [49]. This at least shows that people with lower incomes may prefer cheaper food. At the same time, many authors note that human SES is one of the most important factors currently being studied that has an influence on consumer food choice [44,45,46,47,48,49,50,51,52,53,54,55,56,57,58,59,60,61,62].

As shown in the example of Switzerland (the sample consisted mainly of Europeans: Swiss, British, French, etc.), there are differences between people who are poorly educated and those with a high level of education, as well as between people with different income levels in terms of level of fish consumption (from 5 to 23%) and vegetables (from 7.7 to 11%) [50]. In turn, low-income consumers are more likely to consume fried foods (from 14.5 to 24.4%), which is partly consistent with the data we presented. A later study (data collected from 2005 to 2012) showed a statistically significant difference in intake levels for vitamin D, dietary fiber, and fast carbohydrates (which does not exclude fast food consumption) [51].

Similar results have been shown for consumer samples from Denmark [53]. It was shown that depending on the level of education, the diet of a consumer changes. Thus, in the presence of a higher education, compared to a complete secondary education, the consumption of vegetables and fruits increases by 70–82% (especially among men). A similar situation manifests itself both among women and among men in relation to the consumption of fish (differences from 18 to 26%).

A study of consumer behavior in stores of Queensland, Eastern Australia, also found that when shopping, respondents from lower socio-economic groups were less likely to buy foods high in fiber and low in fat, salt, and sugar [54]. Populations with lower incomes purchased fewer types of fresh fruits and vegetables with less frequency than their counterparts in higher-income households. It can be assumed that these categories of food products are a kind of reference markers (socio-economic indicators) that one should pay attention to when designing a study. The research on consumer preferences taking SES into account, also for Australian consumers, has shown that men with a lower SES are more likely to consume tropical fruits (apparently, this type of food can be attributed to the specific features of countries in warm climates, such as Australia) and canned fish [55]. For the latter product, our data also points to increased consumption of canned food, including canned fish, among Russia’s poor consumers.

Research conducted in Holland showed that persons with a low level of education had relatively more kcal from saturated fat (14.5% for the former and 13.8% of energy for consumers with a higher level of education). As authors have noted, these differences can be explained by an higher overall fat intake, as well as higher meat consumption (by 11–14%). It was also discovered that people with higher education consumed relatively more lean meat and low-fat dairy products [52].

In general, the relationship between SES and diet patterns has been demonstrated in other countries as well, for example, samples of consumers from Norway, Canada, the USA, France, Ireland, and other countries of the European Union [56,57,58,59,60,61,62]. This is primarily due to an increase in calories (including due to an increase in the share of consumption of fats and fast carbohydrates, which does not exclude the increased consumption of semi-finished products and canned food), as well as a decrease in the share of consumption of fresh fruits and vegetables in the diets of consumers from these countries when their income falls. In general, the results presented in this chapter indicate the presence of a global trend associated with a change in food consumption depending on the social well-being of respondents.

## 5. Conclusions

In the course of our study, a clear relationship was revealed between the imbalance of the diet and the economic disadvantage of consumers. The presented data and the algorithm for determining the antiradical status of consumers can be useful in studies in other countries of the world. And in our opinion, it is impossible to conduct research on the balance of diets without taking into account the socio-economic conditions of human life. The findings partly explain why the goals set by the WHO to reduce the spread of some food-associated diseases are not being achieved so quickly and sometimes with difficulty. We do not exclude the possibility of using neural networks for these purposes, as this approach will allow researchers to at least compare the values obtained from studies with different designs.

Our data show that even if we do not take into account estimates that fall into the “poorly balanced” range, approximately 36% of Russians still have an unsatisfactory antiradical status, which directly depends on their diet. No less important is the fact that when living in unfavorable conditions or when consuming xenobiotics, the proportion of such people can be increased under different conditions by another ≈10–20%.

## Figures and Tables

**Figure 1 foods-12-02619-f001:**
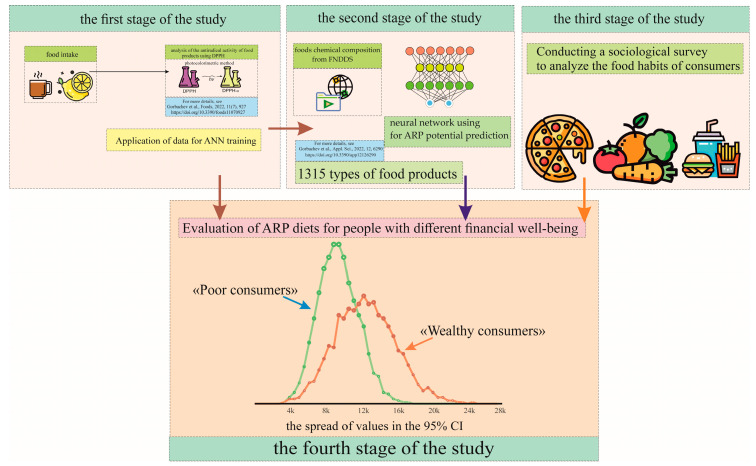
A generalized scheme of a social study on the relationship between consumer well-being and ARP, which involves three preliminary stages of work: a laboratory study of the ARP of food products [17]; the use of an artificial neural network (ANN) to predict ARP based on the chemical composition of food products [25]; and sociological study of consumer preferences.

**Figure 2 foods-12-02619-f002:**
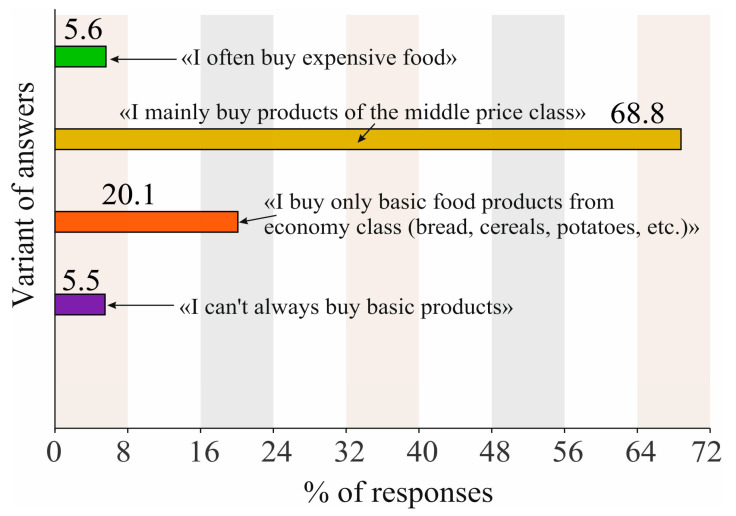
Percentage of surveyed respondents according to their preferences to purchase products from different price categories (Appendix B, question number 14). Note: numbers above the columns show the percentage of consumers who gave corresponding answers.

**Figure 3 foods-12-02619-f003:**
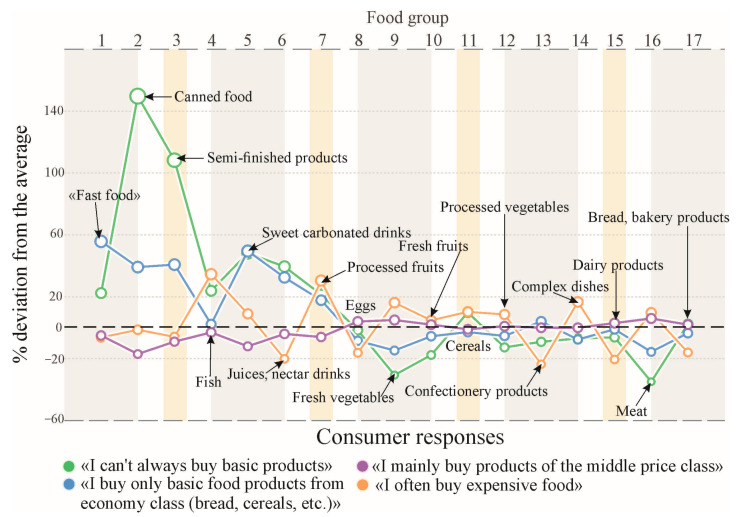
Influence of consumer welfare on the probability of their diets deviating from the population averages. Note: the numbering of product categories in the figure is the same as in Table 1; the average value is shown as a dashed line at 100% (denoted as zero).

**Figure 4 foods-12-02619-f004:**
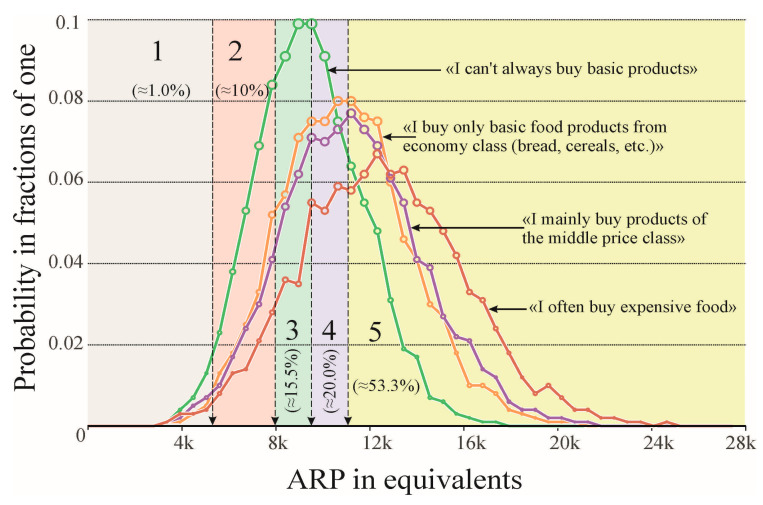
The results of bootstrapping estimates of the ARP of people’s diets based on the data of sociological surveys and the average predictive values obtained using the ANN and laboratory estimates for each of the food groups. Note: ranges of values (marked with numbers from 1 to 5) of the restorative balance of diets are highlighted in color. In parentheses are the percentages of consumers in terms of the entire sample, whose diets fall into one or another range.

**Table 1 foods-12-02619-t001:** Proportions of food groups (essentially the weighted average probability of a food being in the diet on any random day), ranked by increasing values.

No.	Name of the Food Group	Shares in the Diet, in 95% CI
1	Fast food products (bistros, eateries)	1.58–2.61%
2	Canned foods	1.46–4.67%
3	Fast food	1.60–3.76%
4	Fish, fish products	2.73–3.73%
5	Sugary carbonated drinks	2.27–3.70%
6	Juices	2.20–3.65%
7	Processed fruits	3.34–4.42%
8	Eggs	5.43–6.50%
9	Fresh vegetables	4.99–7.78%
10	Fresh fruits	6.18–7.59%
11	Cereals	7.57–8.61%
12	Processed vegetables	7.14–8.56%
13	Confectionery	6.67–8.66%
14	Complex dishes (soups, casseroles)	7.57–9.46%
15	Milk, dairy products	7.26–8.91%
16	Meat, meat products	7.44–11.73%
17	Bread, bakery products	9.30–11.02%
18	Total: 100%	

**Table 2 foods-12-02619-t002:** Estimates of the proportion of participants (consumers) in terms of the entire sample whose values are below the recommended level of consumption of each type of food component, subject to the maximum diversity of dietary choices.

No.	Nutritional Component	Average Recommended Level in Russia	FNDDS *
1	Retinol (μg)	900.0	≈59
2	β-carotenes (μg)	5000.0	≈38
3	Thiamine (B1) (mg)	1.5	1
4	Riboflavin (B2) (mg)	1.8	0.2
5	Niacin (B3) (mg)	20.0	≈1
6	Pyridoxine (B6) (mg)	2.0	≈1
7	Folates (B9) (μg)	400.0	≈5
8	Cobalamin (B12) (μg)	3.0	0.5
9	Vitamin C (mg)	90.0	6
10	Vitamin (D2 + 3) (μg)	12.5	≈73
11	Vitamin E (α) (mg)	15.0	13

*—% of consumers with consumption values below the recommended level when using the chemical composition of the FNDDS database and results of a sociological survey for calculation.

## Data Availability

All data are presented within the article.

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
