# Peer review of "The Impact of Social Well-Being on Population Diet Nutritional Value and Antiradical Status"

_foods, 2023, doi:10.3390/foods12132619_

Round 1

Reviewer 1 Report

Overall this is a good manuscript that needs some adjustments.

Representativeness of the sample needs to be established. The presentation of how data were obtained was a bit complex and left doubts regarding generalizability of the findings.

To provide a broader impact of the research, it is important to address more directly key implications for policy and practice.

Several statistical procedures are applied, without providing much background on why they are appropriate. Also, it is important to provide evidence about the extent to which assumptions underlying these procedures were satisfied.

Some editing will be required for the manuscript. For example, “catering” does not seem to be a logical construct and either should be explained clearly or replaced. Also, "Fligner-Killeen" needs to be used instead of the incorrect “Flinger-Killen.”

As a recommendation, consider changing the title to something like “The impact of social well-being on population diet nutritional value and antiradical status,” which seems a more direct representation of the import of the manuscript.

English usage generally is good and the main points are clear enough, although careful copyediting is needed. Some specific points are addressed in notes to the authors.

Author Response

Dear reviewer!
We are grateful to you for your attention to our manuscript.

  1. Representativeness of the sample needs to be established. The presentation of how data were obtained was a bit complex and left doubts regarding generalizability of the findings.

Dear reviewer, we have added a questionnaire form for the survey as additional materials to the manuscript. Since the original version was in Russian, we made the closest translation. We have also added statistics on the representativeness of the sample and our reasoning to the section "materials and methods" in the manuscript. In general, we are grateful for this remark and took it into account.

  1. To provide a broader impact of the research, it is important to address more directly key implications for policy and practice.
  2. Several statistical procedures are applied, without providing much background on why they are appropriate. Also, it is important to provide evidence about the extent to which assumptions underlying these procedures were satisfied.

We are grateful to the reviewer for this remark, but we justified the use of these methods in early works. In order to make it more clear to the reader, we added links to the text more often and expanded our reasoning where necessary.

  1. Some editing will be required for the manuscript. For example, “catering” does not seem to be a logical construct and either should be explained clearly or replaced. Also, "Fligner-Killeen" needs to be used instead of the incorrect “Flinger-Killen.”

We have taken this remark into account

  1. As a recommendation, consider changing the title to something like “The impact of social well-being on population diet nutritional value and antiradical status,” which seems a more direct representation of the import of the manuscript.

We are grateful to the reviewer for the comment and changed the name of article, it really seems to us more successful.

  1. Overall this is a good manuscript that needs some adjustments.

We are very grateful to the reviewer for the feedback we received and tried to take into account all his comments.

Reviewer 2 Report

I ask the authors to indicate how the sample size was selected The final sample consists of 1001 respondents whose data were collected by different methods. Indicate whether the final sample is representative? According to what parameters is it stratified? If the sample is not representative, this must be stated in the limitations of the study. Please add the number of respondents to the summary.
Since the amount consumed was not determined, I recommend stating this as a limitation of the study
Ln 242- Why is the moisture value set at exactly 68.3%? Please explain.
How were the questions formulated, the answers to which are shown in Figure 2? Please attach the questionnaire as an Appendix.
I ask the authors to expand the Discussion section to include a comparison of food intake in other countries
Figure 2: Please name the x-axis and y-axis of the graph
Ln 307-309- I ask the authors to explain the sentence - on what basis was the conclusion drawn?
Ln 318- Please harmonize the terms "essential" and "basic" foodstuffs. The terms are mixed in the text and in the diagrams.
Table 2- Please provide reference for Average recommended level in Russia.
The strengths and weaknesses of the study are not highlighted in the paper, so they should be included.

I ask the authors to indicate how the sample size was selected The final sample consists of 1001 respondents whose data were collected by different methods. Indicate whether the final sample is representative? According to what parameters is it stratified? If the sample is not representative, this must be stated in the limitations of the study. Please add the number of respondents to the summary.
Since the amount consumed was not determined, I recommend stating this as a limitation of the study
Ln 242- Why is the moisture value set at exactly 68.3%? Please explain.
How were the questions formulated, the answers to which are shown in Figure 2? Please attach the questionnaire as an Appendix.
I ask the authors to expand the Discussion section to include a comparison of food intake in other countries
Figure 2: Please name the x-axis and y-axis of the graph
Ln 307-309- I ask the authors to explain the sentence - on what basis was the conclusion drawn?
Ln 318- Please harmonize the terms "essential" and "basic" foodstuffs. The terms are mixed in the text and in the diagrams.
Table 2- Please provide reference for Average recommended level in Russia.
The strengths and weaknesses of the study are not highlighted in the paper, so they should be included.

Author Response

Dear reviewer!
We are grateful to you for your attention to our manuscript.

  1. I ask the authors to indicate how the sample size was selected. The final sample consists of 1001 respondents whose data were collected by different methods. Indicate whether the final sample is representative? According to what parameters is it stratified? If the sample is not representative, this must be stated in the limitations of the study. Please add the number of respondents to the summary.

- We are grateful to the reviewer for this remark, we have expanded the chapter "Materials and methods". The sample we received was indeed collected by different methods, the reviewer is right here. This is dictated by objective necessity. 1. Very old age groups could not be included in the study – the fact is that in rural areas, as well as older people (over 70 years old) they do not have sufficient computer literacy and would not otherwise be able to participate in the study.  And because of this reason we use different methods. We added this to the text of the study.

Margin Error estimation data for our sample has also been added to the text.

  1. Since the amount consumed was not determined, I recommend stating this as a limitation of the study.

- We added this to the text, and took into account the remark at the end of the manuscript.

  1. Ln 242- Why is the moisture value set at exactly 68.3%? Please explain.

-We added this to the text, and took into account. This value was taken from the FNDDS database as the average value for our samples.

  1. How were the questions formulated, the answers to which are shown in Figure 2? Please attach the questionnaire as an Appendix.

- We thank the reviewer we took this remark into account. We add Appendix B (question number 14).

  1. I ask the authors to expand the Discussion section to include a comparison of food intake in other countries

- We thank the reviewer; we took this remark into account. We have expanded the relevant reasoning in the text and made comparisons with other data published earlier by other authors.

  1. Figure 2: Please name the x-axis and y-axis of the graph

- We thank the reviewer we took this remark into account. For figure 2 and 4.

  1. Ln 307-309- I ask the authors to explain the sentence - on what basis was the conclusion drawn?

-We thank the reviewer we took this remark into account.

We have softened the words in the text and indicated that this is an assumption that was intended taking into account the conclusions presented in the text later.

8. Ln 318- Please harmonize the terms "essential" and "basic" foodstuffs. The terms are mixed in the text and in the diagrams.

- We thank the reviewer; we took this remark into account. These are the inaccuracies of our translation; we have changed the text, in accordance with Figure 3.

  1. Table 2- Please provide reference for Average recommended level in Russia.

- We thank the reviewer; we took this remark into account, we brought it earlier in another work, but we add this reference once more.

  1. The strengths and weaknesses of the study are not highlighted in the paper, so they should be included.

- We thank the reviewer; we took this remark into account.

Reviewer 3 Report

Thank you for involving me in the review of this manuscript. the paper presents the result of assessing the antiradical status of consumers in Russia in relation to their well-being. 

Abstract: The authors write "From the obtained values it fol- 35 lows that consumers with low levels of ARA in their diets are at risk with an increased likelihood 36 of a number of "excess" diseases (diseases of the cardiovascular system, obesity, etc.)." One can speak of risk if the study design allows causal inference, otherwise one speaks of association or probability of an event. Please revise the sentence and specify the study design in the abstract.

which dietary questionnaire was used? is it a previously validated questionnaire in the same population sin examination? please specify

Please provide a flowchart of the sample under study, specifying the inclusion and exclusion criteria.

lines 179-233. Move all content as supplementary material.

Statistical methodology is poor.

The study limitations section is missing.

The English language used is too conversational to be a scientific paper. Please, revise.

Author Response

Dear reviewer!
We are grateful to you for your attention to our manuscript.

  1. Thank you for involving me in the review of this manuscript. The paper presents the result of assessing the antiradical status of consumers in Russia in relation to their well-being. 

- The authors of the manuscript also thank the anonymous reviewer and editor of «Food» journal for the informative and constructive comments that helped make the article more interesting for the reader.

  1. Abstract: The authors write "From the obtained values it follows that consumers with low levels of ARA in their diets are at risk with an increased likelihood 36 of a number of "excess" diseases (diseases of the cardiovascular system, obesity, etc.)."

- One can speak of risk if the study design allows causal inference, otherwise one speaks of association or probability of an event. Please revise the sentence and specify the study design in the abstract.

We thank the reviewer; we took this remark into account.

  1. Which dietary questionnaire was used? is it a previously validated questionnaire in the same population sin examination? please specify

- We thank the reviewer for an important question. We studied the anti-radical status in Russia for the first time and we did not have questionnaires similar to us or any analogues, so we compiled the questionnaire ourselves.

  1. Please provide a flowchart of the sample under study, specifying the inclusion and exclusion criteria.

- We used only two parameters as inclusion and exclusion criteria; 1. this is the age of the participants, it had to be over 18 years old; 2. the criterion of residence on the territory of Russia as a permanent citizen of this country. Participants were instructed about both criteria in advance, so we removed only 1 participant - it turned out to be a student of the school less than 18 years old. For this reason, we have not compiled a detailed flowchart, because it has no illustrative power, and for this reason we have depicted Figure 1 as a flowchart of the stages of the study.

We are grateful to the reviewers for such a detailed acquaintance and analysis of our manuscript.

  1. lines 179-233. Move all content as supplementary material.

- We took this remark into account. We add full list of products in Appendix A.

  1. Statistical methodology is poor. The study limitations section is missing.

- We took this remark into account, and add limitations in the text. As for statistics, we previously described statistical approaches in detail (there are three links in the text), in order to make our article more readable for readers, we removed some detailed descriptions.  We thank the reviewer once again for the time he/she gave us.

Round 2

Reviewer 1 Report

The authors have been responsive to the major concerns of the reviewers.

The quality of expression is high in general. Remaining issues should be addressed through relatively minor copyediting.

Reviewer 3 Report

Manuscript well improved